

# Microbial communities associated with mounds of the Orange-footed scrubfowl *Megapodius reinwardt*

Karla Cardenas Gomez, Alea Rose, Karen Susanne Gibb and Keith A. Christian

Research Institute for the Environment and Livelihoods, Charles Darwin University, Darwin, Northern Territory, Australia

## ABSTRACT

*Megapodius reinwardt*, the orange-footed scrubfowl, belongs to a small family of birds that inhabits the Indo-Australian region. Megapodes are unique in incubating their eggs in mounds using heat from microbial decomposition of organic materials and solar radiation. Little is known about the microorganisms involved in the decomposition of organic matter in mounds. To determine the source of microbes in the mounds, we used 16S and 18S rRNA gene sequencing to characterize the microbial communities of mound soil, adjacent soil and scrubfowl faeces. We found that the microbial communities of scrubfowl faeces were substantially different from those of the mounds and surrounding soils, suggesting that scrubfowls probably do not use their faeces to inoculate their mounds although a few microbial sequence variants were present in both faeces and mound samples. Further, the mound microbial community structure was significantly different to the adjacent soils. For example, mounds had a high relative abundance of sequence variants belonging to *Thermomonosporaceae*, a thermophilic soil bacteria family able to degrade cellulose from plant residues. It is not clear whether members of *Thermomonosporaceae* disproportionately contribute to the generation of heat in the mound, or whether they simply thrive in the warm mound environment created by the metabolic activity of the mound microbial community. The lack of clarity in the literature between designations of heat-producing (thermogenic) and heat-thriving (thermophilic) microbes poses a challenge to understanding the role of specific bacteria and fungi in incubation.

# INTRODUCTION

The orange-footed scrubfowl (*Megapodius reinwardt*) belongs to the small family of birds *Megapodiidae* that inhabits tropical and subtropical monsoon rainforests, mangrove forests, beach forests, and savannah woodlands of the Indo-Australian region (*Jones & Göth, 2008*; *Jones, Dekker & Roselaar, 1995*; *Monk, De Fretes & Reksodiharjo-Lilley, 1997*; *Pattiselanno & Arobaya, 2014*; *Sinclair, 2002*; *White, 1998*). Unlike other birds, members of this family incubate their eggs by burying them in soil, often associated with mounds of

Corresponding author
Keith A. Christian,
Keith.Christian@cdu.edu.au

decaying organic material (*Frith, 1956*; *Jones & Göth, 2008*; *Sinclair, 2002*), beach sands or volcanically heated soils (*Crome & Brown, 1979*; *Frith, 1956*).

In Australia, the orange-footed scrubfowl has adapted to live successfully among humans in urban areas where scrubby vegetation is abundant (*Jones, 2014*). Their mounds are volcano shaped structures of various sizes but reaching heights over 3 m and weights of five tonnes (*Banfield, 1912*; *Harris et al., 2014*). Their composition depends upon location: near beaches, mounds tend to have a high proportion of sand mixed with a small proportion of leaf litter (*Palmer, Christian & Fisher, 2000*; *Wiles & Conry, 2001*); in areas with heavier clay-type soils, mounds have a higher volume of leaf litter and a small proportion of soil (*Crome & Brown, 1979*).

After eggs are laid inside chambers up to half a metre within the mounds (*Harris et al., 2014*), the female checks the temperature and then covers the mound with abundant leaf litter (*Crome & Brown, 1979*; *Imansyah et al., 2009*). Mound maintenance is shared between the female and male, occasionally several pairs use a single mound simultaneously (*Crome & Brown, 1979*; *Jones, Dekker & Roselaar, 1995*). The parents return to maintain the same mound over several years by adding new materials and increasing the size of the mound (*Crome & Brown, 1979*; *Frith, 1956*; *Palmer, Christian & Fisher, 2000*). Some mounds have been recorded in continual use over 40 years (*Banfield, 1912*; *Jones, Dekker & Roselaar, 1995*; *Palmer, Christian & Fisher, 2000*). Twelve to thirteen eggs are laid individually at intervals of approximately 13 days during a breeding season (*Crome & Brown, 1979*).

The heat required for incubation in the mounds results from a combination of solar radiation and microbial decomposition of organic materials (*Frith, 1956*; *Harris et al., 2014*; *Jones & Birks, 1992*; *Jones, 2014*; *Palmer, Christian & Fisher, 2000*; *Pattiselanno & Arobaya, 2014*; *Seymour & Ackerman, 1980*; *Sinclair, 2002*). Adequate levels of moisture are essential for microbial activity, however, excess water can result in a quick decomposition of leaf litter and a subsequent mound collapse (*Booth & Seymour, 1984*; *Frith, 1956*; *Palmer, Christian & Fisher, 2000*; *Seymour, 1995*; *Seymour & Bradford, 1992*). Factors driven by microbial activity such as temperature and gas levels play a pivotal role during the incubation period (*Seymour & Bradford, 1992*).

Temperatures exceeding 70 °C can be produced by microbes in organic mulches, hay, and manure in a process historically termed "microbial thermogenesis" (*Bartholomew & Norman, 1953*; *Brock, 2012*; *Wedberg & Rettger, 1941*). Microbes can be categorised as being mesophilic (thriving in moderately warm temperatures) or thermophilic (thriving in extremely warm environments, including up to ~70 °C) (*Beffa et al., 1996*), but the latter term is sometimes used more broadly to include the mesophilic temperature range. Little is known about the species of microorganisms involved in the generation of heat associated with organic material decomposition in scrubfowl mounds, and it is unclear how microbial communities related to incubation become established in the mound. The generation of heat by microbes in the mound may simply be a by-product of general microbial decomposition, or it may be dependent on the presence of specialist thermogenic microorganisms.

If specialist heat-producing microbes are required for the generation of heat, then this raises questions about the source of these microbes and how a new mound becomes inoculated with the appropriate microbes. Animals exchange microorganisms with their immediate surroundings, including nests, hollows and burrows (*Brandl et al., 2014*; *Goodenough & Stallwood, 2012*; *Koller, Dworschak & Abed-Navandi, 2006*). Bacterial assemblages in nest materials of the reed warbler (*Acrocephalus scirpaceus*) have some overlaps regarding bacteria species with nestling faeces, suggesting that a certain amount of bacterial transmission from bird faeces to nesting materials is likey to occur (*Brandl et al., 2014*). Alternatively, the source of microbes in the mound may simply be the soil that the birds routinely scratch onto the mound from the surrounding shallow soil.

To investigate the source of microbes in the mounds we used 16S and 18S rRNA gene sequencing to describe the bacteria and fungi communities associated with mounds, soil around mounds and scrubfowl faeces. We tested two hypotheses related to the source of microbes in the mound: (1) scrubfowls inoculate their mounds with thermogenic microbes found in their faeces, and (2) the microbial composition of mounds is similar to that of the surrounding shallow soil.

## MATERIALS AND METHODS

### Types of samples, locations and sampling scheme

The samples were collected from the vicinity of four different mounds (labeled as A, B, C and D) located in the urban area of Darwin, Northern Territory, Australia on a single day in February 2019. The sampling scheme and collection method were approved by the Animal Ethics Committee of Charles Darwin University (project number: A19002) and by the Parks and Wildlife Commission of the Northern Territory, Australia (field study approval number: 64780). The four mounds were located within 1.0 km of each other. Two mounds (A and B) were located on the Charles Darwin University campus (<200 m apart), and the other two (<200 m apart) were located in an adjacent residental area. The samples (approximately 50 g) were collected <24 h prior to DNA extraction. We collected four types of samples: mound soil, deep soil, shallow soil and scrubfowl faeces. Mound soil was collected at a depth of 30 cm to replicate the depth of holes commonly dug in the mounds by birds, and to minimize litter or surface soil contamination (three samples per mound, thus $N = 12$), shallow soil was collected 2 m from the mounds at a depth of approximately 2–5 cm (3 samples per mound, thus $N = 12$), deep soil was only collected 2 m from mound C from a depth of 30 cm ($N = 3$) and scrubfowl faeces were collected from footpaths to reduce the mixing of faeces and soil ($N = 6$). The replicate samples were extracted, sequenced and analysed individually. Although the faecal samples were collected near mounds A, B and C, it was not possible to determine if these samples originated from the birds associated with the respective mounds.

### DNA extraction

DNA was extracted from samples using the Qiagen DNeasy PowerSoil Kit (Qiagen, Valencia, CA, USA), following the manufacturers protocol without modifications.

## 16S and 18S rRNA sequencing and data processing

We sent ten nanograms of DNA per sample to the Australian Centre for Ecogenomics (ACE) for sequencing the 16S and 18S rRNA genes on an Illumina MiSeq following the manufacturer's guidelines. Primers 515-F (GTGYCAGCMGCCGCGGTAA) and 806-R (GGACTACNVGGGTWTCTAAT) (*Apprill et al., 2015*; *Parada, Needham & Fuhrman, 2016*) amplified the 16S rRNA gene, while ITS3-F (GCATCGATGAAGAACGCAGC) and ITS4-R (TCCTCCGCTTATTRATATGC) (*White, Nagarajan & Pop, 2009*) amplified the 18S rRNA gene. ACE processed the sequences to sequence variants (SVs) using the following pipeline. Poor quality sequences (<15 bases) were removed from the reads with Trimmomatic software (*Bolger, Lohse & Usadel, 2014*), then hard trimmed to 250 bases. Reads were converted to SVs using the QIIME-2 workflow with default parameters and the DADA-2 algorithm (*Callahan et al., 2016*; *Caporaso et al., 2010*). SVs were assigned taxonomy through BLAST+ using the reference databases SILVA (*Quast et al., 2013*) for 16S, and UNITE (*Kõljalg et al., 2013*) for the 18S data. SVs were removed if they occurred in <1% of samples or contained fewer sequences than 0.01% of the total sequence abundance. Additionally, sequences were excluded if they were not classified as bacteria or Archaea (16S dataset) or fungi (18S dataset). All sequences were rarified to the lowest common sequence number per sample using a threshold of 10,000 for the 16S and 5,000 for the 18S datasets. Similar to (*Epstein et al., 2021*), we compared the nMDS outputs for the rarefied and unrarefied datasets and found that rarefying did not change the output (*e.g.* 16S stress of unrarefied data = 0.11 compared to 0.1 for rarefied data). Rarification resulted in the removal of two faecal samples and one sample from mound soil from the 16S dataset, and three faecal samples from the 18S. Raw data files in FASTQ format were deposited in the NBCI Sequence Read Archive under BioProject ID PRJNA806506 and project information is accessible through this NCBI project link.

## Data analysis

The two datasets (16S for bacterial and 18S for fungal taxa) were analysed in R (version 3.2.2) (*R Core Team, 2017*) and Primer -7 (*Clarke & Gorley, 2015*) by permutational MANOVA (PERMANOVA; 999 permutations) with the sampled mound locations (A, B, C and D) as the fixed factors within the four sample categories (faeces, soil from mounds, shallow and deep soil). Alpha diversity was examined between the four sample categories and a Bray Curtis distance matrix was calculated and visualized by a nonmetric multidimensional scaling (nMDS) graph using the *phyloseq* package (*McMurdie & Holmes, 2013*). Bacterial and fungal phyla with a relative abundance higher than 1% across all samples were viewed in taxa plots. For both fungal and bacterial communities, we calculated the contribution of each species (%) to the dissimilarity between mound and shallow soil (SIMPER analysis) and also compared samples pairwise. The relative abundances of bacterial and fungal taxa for each sample category and location were visualised in shade plots.

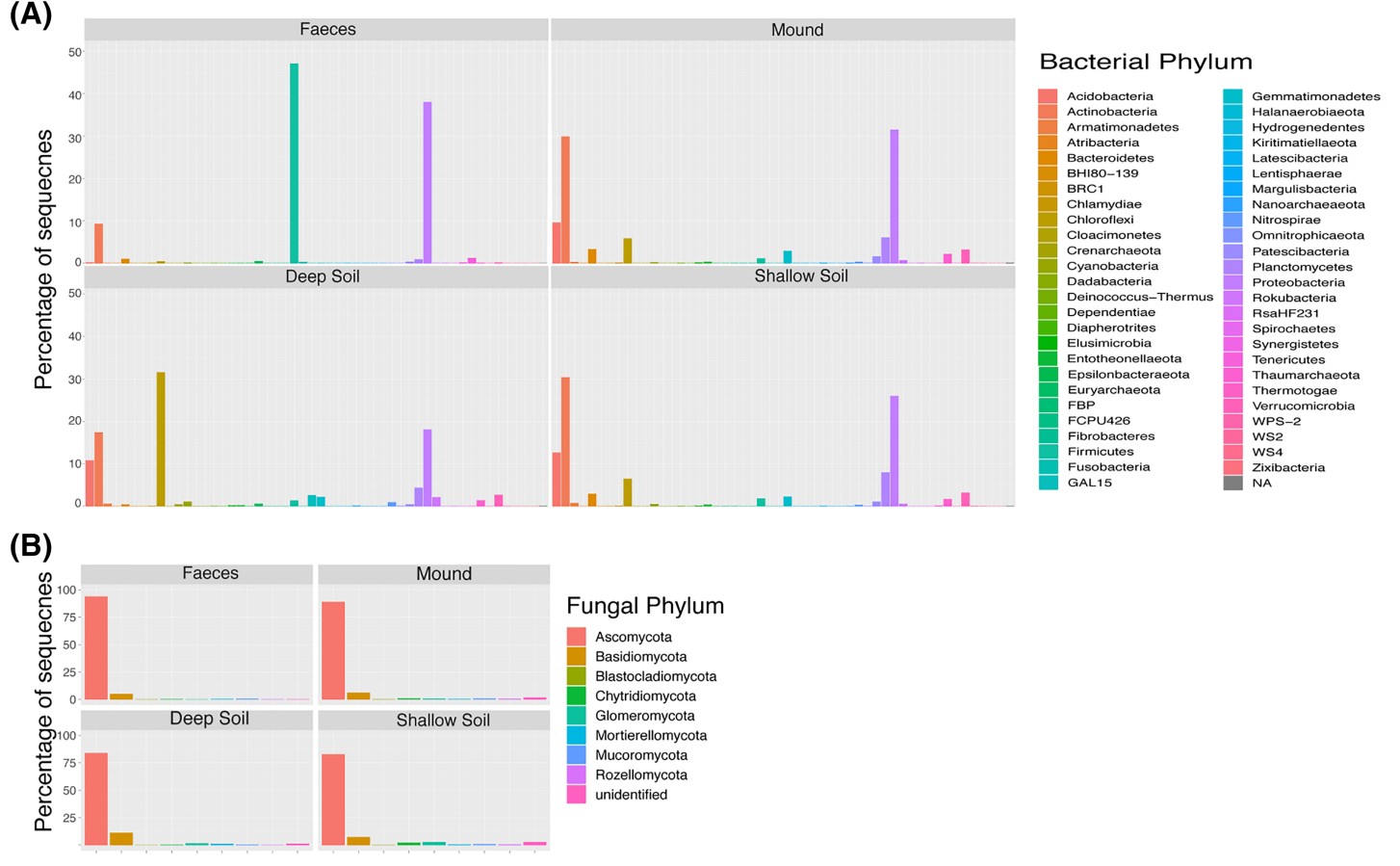

**Figure 1 Taxa plot.** Percentage of contribution of each bacterial (A) and fungal (B) phylum for each sample category (mound, shallow soil, deep soil, faeces). Taxa shown are those that contributed to greater than one percent of all samples.

## RESULTS

### Microbial composition and diversity

Microbial composition of the mounds is broadly illustrated at the level of phyla in Fig. 1. A total of 17,736 bacterial and 1,887 fungal SVs were identified in this study. The bacterial and fungal alpha diversity for mound soil, shallow soil, deep soil and faeces in every mound location is represented by the Shannon index (Table 1). The highest bacteria diversity was in shallow soil near mound C (7.58 ± 0.18) and the lowest in faeces from the same location (3.30 ± 0.75). The highest fungal alpha diversity was in shallow soil near mound A (4.83 ± 0.36), whereas the lowest was in deep soil near mound C (2.36 ± 1.76). Alpha diversity and P values across sample categories are shown in Tables S1 and S2, Fig. S1.

### Comparisons of microbial communities

The sample sizes for mound soil and shallow soil were sufficiently large to allow a statistical comparison of the replicates of these soil categories. "PermDISP" is a permutational distance-based test for homogeneity among the replicates within the locations, and this statistic was not significantly different among replicate samples of mound soil for either bacteria ($p = 0.07$) or fungi ($p = 0.99$) (Table S3). However, there was a significant

**Table 1 Shannon's alpha diversity.**

| Sample | Bacteria mean ± SD (N) | Fungi mean ± SD (N) |
|---|---|---|
| Mound_A | 6.40 ± 0.37 (3) | 4.23 ± 1.05 (3) |
| Mound_B | 6.35 ± 0.27 (3) | 4.30 ± 1.03 (3) |
| Mound_C | 5.97 ± 1.52 (3) | 3.49 ± 1.46 (3) |
| Mound_D | 7.36 ± 0.23 (3) | 4.06 ± 0.44 (3) |
| Shallow_Soil_A | 6.02 ± 0.04 (3) | 4.83 ± 0.36 (3) |
| Shallow_Soil_B | 6.22 ± 0.12 (3) | 4.00 ± 1.31 (3) |
| Shallow_Soil_C | 7.58 ± 0.18 (3) | 3.97 ± 0.55 (3) |
| Shallow_Soil_D | 6.67 ± 0.25 (3) | 4.41 ± 0.37 (3) |
| Deep_Soil_C | 6.64 ± 0.12 (3) | 2.36 ± 1.76 (3) |
| Faeces_A | 4.63 ± 0.38 (2) | 3.21 ± 0.43 (2) |
| Faeces_B | 3.50 ± − (1) | 2.79 ± − (1) |
| Faeces_C | 3.30 ± 0.75 (3) | 2.79 ± 0.57 (3) |

Note:
Average (±one standard deviation) Shannon's alpha diversity for the bacterial and fungal taxa sampled at each mound location (labelled A–D) from each sample category (mound, shallow soil, deep soil, faeces). SD = standard deviation.

difference in microbial communities among replicate samples of shallow soil for both bacteria ($p < 0.01$) and fungi ($p < 0.001$) (Table S4).

An nMDS (Fig. 2) illustrates how bacterial and fungal communities differed between mounds A–D for faeces, soil from the mounds and shallow and deep soils. Bacterial and fungal communities of faeces did not cluster with any other type of sample in any mound location. The PERMANOVA showed that there were significant differences among all sample categories and locations for both fungal and bacterial communities ($p = 0.001$ for both bacterial and fungal samples; Tables 2 and 3). Given that the faecal samples did not cluster with any other sample category, this type of sample was excluded from further statistical analysis including the pairwise tests.

The shade plots (Figs. 3 and 4) show the microbial composition at the lowest possible taxonomic resolution in each sample category and mound location. Bacteria communities of soil from mounds, deep and shallow soils had substantial overlaps in bacteria composition, but these soil categories had much less overlap with the faecal samples (Fig. 3). The SVs with the highest relative abundance among the soil categories belonged to the bacteria families *Solirubrobacteraceae*, *Acidothermaceae*, *Bacillaceae* and *Nitrososphaeraceae*. Soils from the four mounds had a relatively high abundance of SVs belonging to the bacteria family *Thermomonosporaceae*, SVs from this family were in lower quantities in shallow and deep soils. SVs belonging to the family *Pyrinomonadaceae* had a relatively high abundance in shallow soil and were absent in the rest of sample categories. Deep soil contained a high abundance of SVs belonging to uncultured *Choloflexi* and *Gaiellales* families. Faeces contained a relatively high abundance of SVs of the bacteria families *Enterobacteriaceae*, *Peptostreptococcaceae* and *Diplorickettsiaceae*, SVs from these families were absent in the other sample categories.

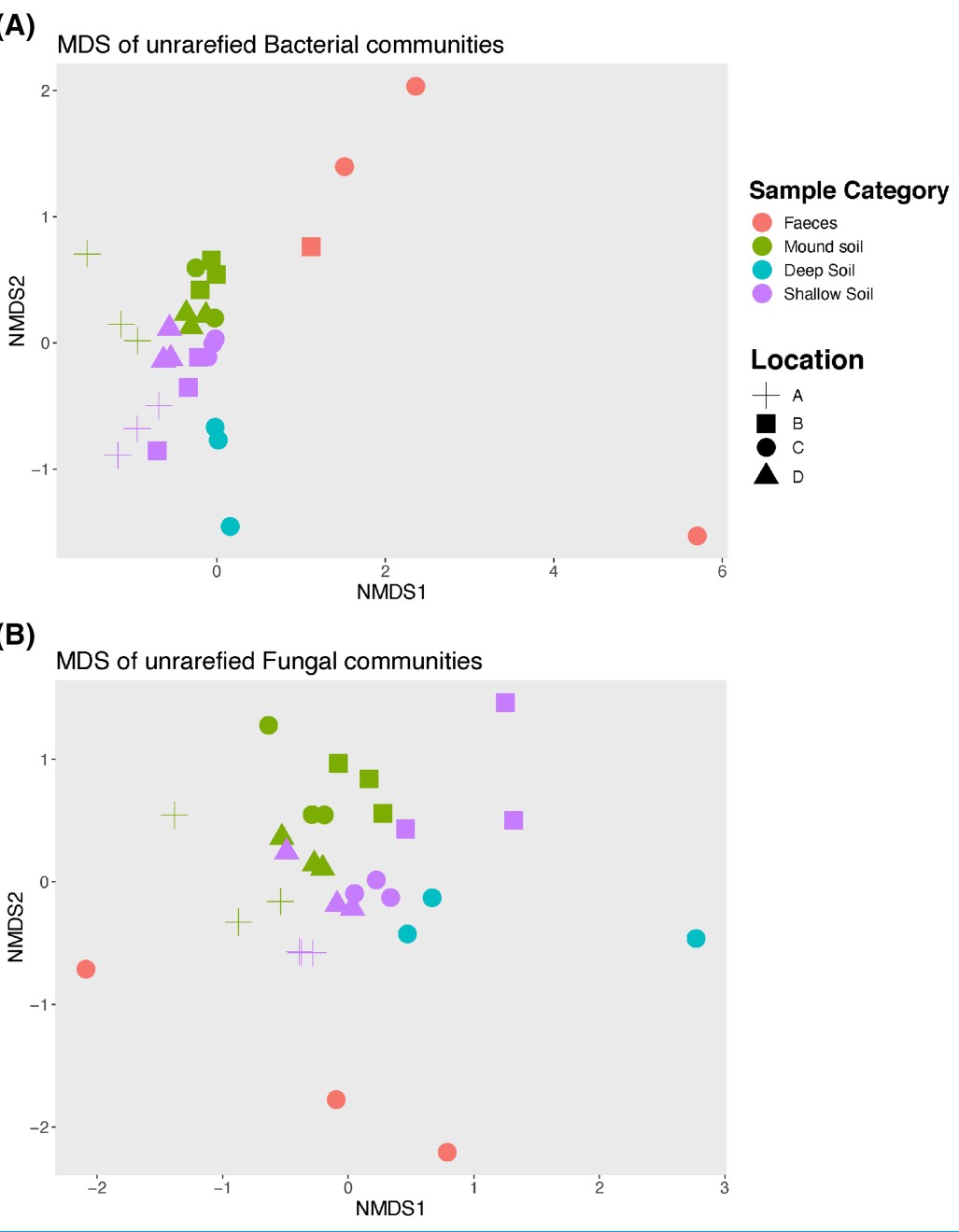

**Figure 2 Non-metric Multidimensional scaling (nMDS) graph.** Unrarefied bacterial (A) and fungal (B) communities sampled at each mound location (labelled A–D) from each sample category (mound, shallow soil, deep soil, faeces). Bacterial community stress = 0.1, and fungal community stress = 0.18. Number of dimensions for each nMDS graph = 2, *N* = 30.

Fungal communities of mounds and adjacent shallow soils overlaped at all mound locations (Fig. 4). SVs belonging to the fungal families *Aspergillaceae, Nectriaceae, Didymosphaeriaceae, Chaetomiaceae, Savoryellaceae* and the order Pleosporales had a relatively high abundance in these two sample categories. Fungal communities in deep soil

**Table 2 PERMANOVA analysis testing differences in bacterial taxa between sample categories.**

**a) Mains test**

| Factor PERMANOVA | Pseudo-F (df) | ECV | *P* value | PermDISP *P* value |
|---|---|---|---|---|
| Location | 3.4 (3) | 27.8 | 0.001 | >0.05 |
| Sample category (location) | 2.7 (7) | 38.1 | 0.001 | <0.05 |

**b) Pairwise tests**

| Groups | *t* (df) | *P* value | Permutations |
|---|---|---|---|
| Location | | | |
| A, B | 1.9 (8) | <0.01 | 986 |
| A, C | 2.2 (7) | <0.01 | 988 |
| A, D | 2.0 (8) | <0.01 | 985 |
| B, C | 1.6 (7) | <0.01 | 988 |
| B, D | 1.9 (8) | <0.01 | 985 |
| C, D | 1.9 (7) | <0.01 | 977 |
| Sample category | | | |
| Mound, shallow soil | 1.6 (21) | <0.01 | 998 |
| Mound, deep soil | 1.7 (12) | <0.05 | 348 |
| Shallow soil, deep soil | 1.6 (13) | 0.001 | 401 |

Note:
The sample categories (mound, shallow soil, deep soil, faeces) were fixed in the sampled mound locations (labelled A–D). (a) mains test. (b) pairwise tests. "df" degrees of freedom, "ECV" square root of estimates of components of variation indicating the size of the effect due to that factor as average % SV dissimilarity (residual ECV 47.0). *P* value is based on >995 unique permutations; "PermDISP" permutational distance-based test for homogeneity of multivariate dispersions for main factors.

contained a relatively high abundance of SVs belonging to the family *Venturiaceae*, but this was absent in the other sample categories. Faeces contained relatively high abundance of SVs of the fungal families *Pleosporaceae*, *Nectriaceae*, *Aurebasidiaceae* and the orders Hypocreales and Pezizales, SVs from these families and orders had a relatively low abundance in the other sample categories.

# DISCUSSION

We found that the replicate samples of mound soil were homogeneous (Table S3), but the replicate samples of shallow soil were heterogeneous (Table S4). Heterogeneity among soil samples is common, reflecting the complex and extremely variable physical and compositional differences in soil, even among samples taken from close proximity (*Brockman & Murray, 1997*; *Nunan, 2017*). The homogeneity among the samples taken from the mound likely reflects the frequent and thorough mixing of the soil by the birds. We also found that the microbial communities of scrubfowl faeces differed significantly in composition from those of the mounds and surrounding shallow and deep soils. However, faeces and soil from the mounds had slight overlapps in bacterial composition. SVs belonging to the families *Solirubrobacteriaceae*, *Bacillaceae* and *Thermomonosporaceae* that were in high relative abundance in mounds, were in very low relative abundance in faeces. The SVs with the highest relative abundance detected in scrubfowl faeces corresponded to common intestinal microflora of wild birds. Members of

**Table 3 PERMANOVA analysis testing differences in fungal taxa between sample categories.**

**a) Mains test**

| Factor PERMANOVA | Pseudo-F (df) | ECV | *P* value | PermDISP *P* value |
|---|---|---|---|---|
| Location | 2.1 (3) | 22.4 | 0.001 | 0.01 |
| Soil category (Location) | 2.2 (6) | 34.0 | 0.001 | 0.01 |

**b) Pairwise tests**

| Groups | *t* (df) | *P* value | Permutations |
|---|---|---|---|
| Location | | | |
| A, B | 1.9 (10) | <0.01 | 412 |
| A, C | 1.9 (10) | <0.01 | 401 |
| A, D | 1.8 (10) | <0.01 | 395 |
| B, C | 1.6 (10) | <0.01 | 416 |
| B, D | 1.6 (10) | <0.01 | 404 |
| C, D | 1.5 (10) | <0.01 | 401 |
| Sample category | | | |
| Mound, Shallow soil | 1.4 (22) | 0.001 | 994 |
| Mound, Deep soil | 1.3 (13) | <0.01 | 405 |
| Shallow soil, Deep soil | 1.2 (13) | <0.05 | 402 |

**Note:**
The sample categories (mound, shallow soil, deep soil, faeces) were fixed in the sampled mound locations (labelled A–D). (a) mains test. (b) pairwise tests. "df" degrees of freedom, "ECV" square root of estimates of components of variation indicating the size of the effect due to that factor as average % SV dissimilarity (residual ECV 53.8). *P* value is based on >995 unique permutations; "PermDISP" permutational distance-based test for homogeneity of multivariate dispersions for main factors.

*Enterobacteriaceae, Peptostreptococcaceae, Pleosporaceae* and *Aureobasidiaceae* have been previously isolated from the faeces and bills of turkey vultures (*Cathartes aura*) (*González-Braojos et al., 2012*; *Winsor, Bloebaum & Mathewson, 1981*), penguins (*Dewar et al., 2013*), hummingbirds (*Lee et al., 2019*) rooks (*Vlahović et al., 2010*) and beaks and cloacas of Mallard ducks (*Dynowska, Meissner & Pacyńska, 2013*). Although we cannot exclude the possibility that there is some bacteria transmission from the bird faeces to the mound, we do not have direct evidence that they inoculate their mounds with faeces, and any such transmission would only involve a small component of the mound microbiota (Figs. 3 and 4).

With respect to our hypothesis that the microbial composition of mounds is similar to that of the surrounding shallow soil, the PERMANOVA analysis indicated that the structure of the microbial community of the mounds was significantly different to that of the surrounding shallow soil. Nevertheless, the taxa plot (Fig. 1) and shade plots (Figs. 3 and 4) showed that mounds and shallow soils had overlapping microbial composition, indicating that the microbial communities of these sample categories differed in relative abundance rather than in microbial composition. Interestingly, previous studies on microbial communities in nests of mound-building ants demonstrated that bacteria and fungal communities of the nest differed significantly from those of the surrounding soils with respect to both community structure and taxonomic composition (*Lindström, Timonen & Sundström, 2021*; *Lindström et al., 2019*). It is possible that the differences in relative abundance between microbial communities of scrubfowl mounds

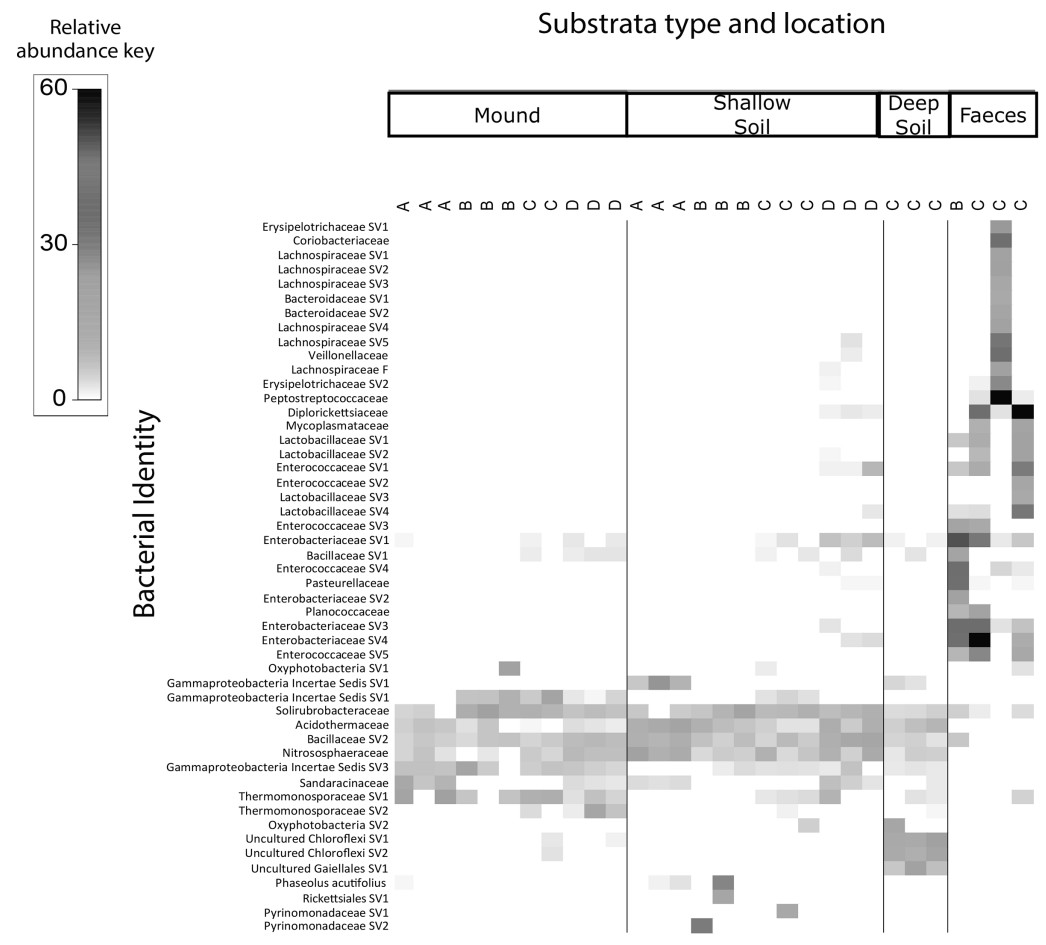

**Figure 3** **Relative abundance heat map of the top 50 most abundant bacterial taxa.** Sampled at each mound location (labelled A–D) from each sample category (mound, shallow soil, deep soil, faeces). Bacterial SVs are assigned to the family where possible, but if family is not resolved, then SVs are assigned to orders.

and surrounding soils are related to the increased temperatures resulting from microbial metabolic activity in the mound. Thus, the source of the microbes in the mound is probably the surrounding soil, but the environmental conditions of the mound favour some microbes over others, resulting in a microbial community that is structurally different from the adjacent soil.

Our description of the microbial communities in the different soil categories is consistent with previous studies of microbes in the enviroment. Members of the families *Solirubrobacteraceae, Acidothermaceae, Bacillaceae, Nitrososphaeraceae, Aspergillaceae* and *Nectriaceae* have been isolated from different types of soil samples, including woody substrata, decaying herbaceous material and soil aggregates (*Furlong et al., 2002*; *Horn, 2003*; *Kerou et al., 2016*; *Kim et al., 2007*; *Klich, 2017*; *Mohagheghi et al., 1986*; *Rempfert et al., 2017*; *Rossman et al., 1999*; *Samuels, 1988*; *Stenfors Arnesen, Fagerlund & Granum, 2008*; *Stieglmeier et al., 2014*), where some species of these families are involved in fundamental roles, including the cycling of organic matter (*Stenfors Arnesen, Fagerlund & Granum, 2008*)

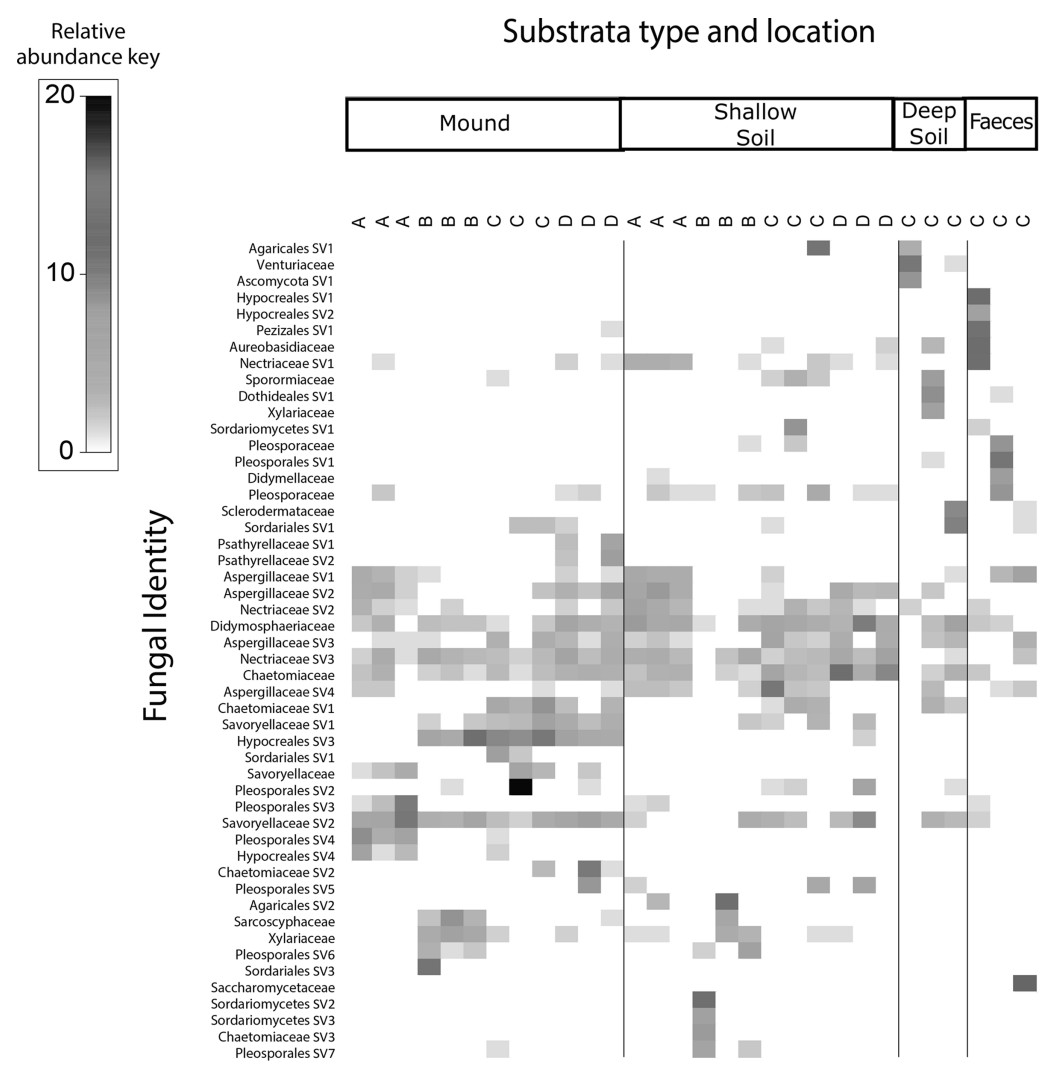

**Figure 4 Relative abundance heat map of the top 50 most abundant fungal taxa.** Sampled at each mound location (labelled A–D) from each sample category (mound, shallow soil, deep soil, faeces). Fungal SVs are assigned to the family where possible, but if family is not resolved, then SVs are assigned to orders.

and nitrogen (*Stieglmeier et al., 2014*). The detection of SVs belonging to the *Sclerodermataceae* and uncultured *Choloflexi* and *Gaiellales* families in deep soil is consistent with previous studies that reported species of these families in hot springs, sediment, lake sediment, geothermal soils (*Yamada & Sekiguchi, 2009*) and subterranean environments including a volcanic cave (*Riquelme et al., 2015*); and as ectomycorrhizals around warm temperate regions (*Jeffries, 1999*). It is therefore unsurprising that these families were highly abundant in deep soil.

Interestingly, only soil from the mounds had a relatively high abundance of SVs belonging to *Thermomonosporaceae*, a thermophilic soil bacterial family with the ability to degrade cellulose from plant residues (*Spiridonov & Wilson, 1998*). Thermomonosporas are frequently isolated from self-heated organic materials, including mouldy hays or urban

wastes (*Kroppensted & M, 2006*; *Lawrence et al., 1986*). It is possible that members of *Thermomonosporaceae* play a role in heat production in mounds, contributing to the incubation of eggs. One important unanswered question is whether the heat generated through decomposition of organic material in the mound is simply a metabolic product of typical soil microbial metabolism, or if specialist thermogenic microbes are involved. The high abundance of the family *Thermomonosporaceae* in mounds but not in the nearby shallow soil seemingly support a model involving specialist microbes. Nevertheless, the cause  and effect relationships are not known because it is not clear whether Thermomonosporas and/or other microbes are disproportionately contributing to the generation of metabolic heat, or if they are simply thermophilic and therefore thrive in the warm environment created by other microbes. The fact that most environmental microbes are difficult to culture (*Bodor et al., 2020*) poses a serious challenge to answering this question.

In some organic mulches, successional stages characterised by different microbial communities result in increasingly higher temperatures. Although scrubfowl eggs benefit from a moderately warm mound environment, they would certainly perish if microbial metabolism resulted in temperatures as high as those that routinely occur in some organic mulch mixtures (*Beffa et al., 1996*). The extent to which the behaviour of the adult birds or the dominance of mesophilic microbes act to maintain moderately warm temperatures in the mound without the production of extreme temperatures is unknown. For there to be thermal conditions appropriate for successful incubation, there must be a delicate balance between external environmental conditions, microbial activity within the mound, and the periodic manipulation and excavation of the mound by adults. The relative roles of each of these factors remain unknown.

## CONCLUSIONS

We found that the microbial communities of scrubfowl feaces were significantly different from those of the surrounding deep and shallow soils, suggesting that scrubfowls probably do not use their faeces to inoculate their mounds, although a small component of the mound microbiota was also found in faecal samples. The microbial communities of the mound differed significantly from those of the surrounding soils in terms of relative abundance. We speculate that the relative high abundance of the family *Thermomonosporceae* in mounds compared to its very low relative abundance in surrounding shallow soils might indicate a model of specialist microbes in the mound. Nevertheless, the extent of specialist microbes involved in the production of heat and the complexities of the heat-producing metabolic pathway in scrubfowl mounds remain unknown. Future studies of fungi and bacteria in mounds and surrounding soils using metagenomics and RNA sequencing might help to identify specific metabolic pathways related to the production of heat and the identification of any specialist microbes involved in the process. Further, it is important to study the role of adult bird activity to facilitate the conditions for incubation and the appropriate levels of heat production inside the mounds.

### Funding

This project was funded by the Charles Darwin University "Rainmaker Scheme". The funders had no role in study design, data collection and analysis, decision to publish, or preparation of the manuscript.

### Grant Disclosures

The following grant information was disclosed by the authors:
Charles Darwin University "Rainmaker Scheme".

### Competing Interests

The authors declare that they have no competing interests.

### Author Contributions

- Karla Cardenas Gomez performed the experiments, analyzed the data, prepared figures and/or tables, authored or reviewed drafts of the article, and approved the final draft.
- Alea Rose analyzed the data, prepared figures and/or tables, authored or reviewed drafts of the article, and approved the final draft.
- Karen Susanne Gibb conceived and designed the experiments, prepared figures and/or tables, authored or reviewed drafts of the article, and approved the final draft.
- Keith A Christian conceived and designed the experiments, performed the experiments, analyzed the data, prepared figures and/or tables, authored or reviewed drafts of the article, collected the samples from the field, and approved the final draft.

### Animal Ethics

The following information was supplied relating to ethical approvals (*i.e.*, approving body and any reference numbers):

The Animal Ethics Committee of Charles Darwin University approved the study (project number: A19002).

### Field Study Permissions

The following information was supplied relating to field study approvals (*i.e.*, approving body and any reference numbers):

Field experiments were approved by the Parks and Wildlife Commission of the Northern Territory, Australia (field study approval number: 64780).

### Data Availability

The sequences are available at GenBank: PRJNA806506.

### Supplemental Information

Supplemental information for this article can be found online at http://dx.doi.org/10.7717/peerj.13600#supplemental-information.

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
