# Peer review of "Microbial communities associated with mounds of the Orange-footed scrubfowl Megapodius reinwardt"

_PeerJ, doi:10.7717/peerj.13600_

## Round 0.1 · original submission · Major Revisions

Both the reviewers and I agree that this is an interesting and well written manuscript. The overall presentation is concise, which is a positive, but I also agree with the reviewers that some more detail in particular would be beneficial, especially with respect to the ecology of nest building in the introduction. And is there any behavioural data providing evidence of deliberate inoculation of the mounds with faeces etc?

I agree with reviewer 2 that although the conclusions seem reasonable, given the approach and sample size it might be sensible to tone down the overall conclusions. From my point of view, dissimilarity between two microbial communities isn't automatic evidence that they are unrelated, as we might expect that only a fraction of the microbes capable of thriving within the avian digestive tract would be capable of thriving in a mound of soil. To get at this we need to know the relative proportion of shared ASVs between samples, and what they are, rather than group level estimates of dissimilarity, though I agree the heat maps and ordinations are useful.

Some methodological detail is missing, such as number of NMDS axes ordinated to, and stress levels.

In addition, there doesn't seem to be any formal analysis of alpha diversity differences. It would be useful to model differences in diversity across samples, even if you have to average pseudoreplicates from each location. Graphing these data would also be useful.

I anticipate you will be able to deal with these revisions fairly easily and look forward to receiving a revised manuscript.


Additional Comments:
L64 "scrubfowl"
L166: 'location'

·

Basic reporting

The manuscript is clear, well written and the English of a high standard. The topic is well introduced and the background literature well covered. The terms ‘thermogenesis’ and ‘thermophilic’ need to be defined and distinguished in the introduction. The terms thermogenic and thermogenesis need to be used with extreme caution in reference to microorganisms - see the general comments above for reasons why. Figures and tables are well presented and are relevant. I did not see any statement where the raw data was available, so I could not confirm that raw data are available as is PeerJ policy.

Experimental design

The experimental design is straightforward and well described. The methods are easily followed. I’m not an expert on the methods used for determining microbe diversity and abundance and the statistical methods used to analyze this type of data. However, I do believe the methods and statistical analysis used are standard for this type of data.

Validity of the findings

The conclusions reached appear well supported by the data.

Additional comments

Specific comments
Lines 72-78: Also maybe mention there are also species that incubate their eggs in volcanically heat sand, and black sand beaches.

Line 91: Are there such things as ‘specialist thermogenic microorganisms’? The use of the term thermogenic implies that their major function in life is generating heat. But because such microorganisms, by definition are so small, individuals at simply not capable of generating sufficient heat to raise their temperature above the ambient environment, their surface area to volume ratio is such that any heat they generate would be rapidly dissipated to the surrounding environment. The only way these microorganisms can raise temperature above ambient is if they are clumped together by their millions, i.e. they combine together to form a ‘cooperative super colony’. As suggest elsewhere these organisms are poor likely to be thermophilic, i.e. heat ‘loving’ or heat ‘tolerant’ as opposed to ‘thermogenic’. To be defined as ‘thermogenic’, which is a energetically expensive process, and therefore must have some other benefit to the organism because of this cost, this benefit must be explained. One possible explanation might be, that it gives these organisms a completive advantage, i.e. by raising the environmental temperature it kills off or excludes other species of microorganism that might also be using the organic plant material as a food source. So here the definition of ‘thermogenic’ and ‘thermophilic’ needs to be defined and the difference between the two pointed out.

Line 274-275: But previously the authors stated that the species distribution was not different between mounds and surrounding shallow soil, but there was a big difference in the relative abundance of different microbe types in these two environments. So this statement should be altered to be consistent with that in the previous text.

Line 279: Consider replacing ‘thermogenesis’ with ‘metabolic heat production’. As stated above, ‘thermogenesis’ has the connotation that heat is being deliberately produced for the sole purpose of raising the temperature of the organism, whereas it is far more likely that the heat is being produced as a simple by-product of routine metabolism in these microorganisms. Specialist heat producing metabolic pathways, the sole purpose of which is to produce heat have been demonstrated in animals (i.e. the H+ leak in the mitochondria of brown adipose tissue of eutherian mammals, and the Ca+ leak in the sarcoplasmic reticulum of the heater organ of billfishes) and plants, the alternative oxidative pathway in ‘heat producing’ plant organs. Until such metabolic pathways are demonstrated in microorganisms, the use of the terms “thermogenic’ and ‘thermogenesis’ should be avoided in reference to microorganisms.

Reviewer 2 ·

Basic reporting

I did enjoy reading this well written and relatively well structured manuscript on an interesting topic.
Regarding the background provided in the introduction, I would have only liked to get a bit more information about the species ecology and in particular what is known about the mound:

- L72-78: Is there anything known about the construction of the mounds? Who builds the mounds, male or female? Does one mound belong to one pair or are others involved in the process? Are there multiple eggs laid in multiple different chambers per mound?

Experimental design

The authors do present an original study well within the scope of the journal. The question from where the material and the microbes in the mound originate seems interesting and relevant, particularly given the unclear role of thermogenic/thermophilic bacteria in these mounds, that the authors highlight. I do however have several more questions and comments regarding the methods:

- One criticism that I have is that I found it hard to follow what the sample sizes really were and how and which were used in the different analyses. I have made some detailed line-by-line comments on this below regarding that I was unclear where the faeces originated from, and that I was somewhat unclear about the rarefaction process and whether the samples excluded there were not used in any other analyses (see also comment on Fig. 2).

- I found it not very clearly described anywhere how the multiple samples collected at some of the locations were dealt with (e.g., 3 soil samples from the same mound). It seems that they were perhaps just combined per mound in the analyses, but that does not seem to be explained anywhere.

- L112-113: Why did the authors choose to sample 4 mounds within 1km?

- L114-115: What time of the year and in what time window were all the samples collected?

-
- L121: Could you tell if it actually originated from the owners of the respective mound? Around which mounds was the faeces collected? Later, in Table 1 I find it mentioned from 3 sites, in Fig2. there seem to be bacterial samples from 2 sites, and fungal samples from 1 site. This is somewhat confusing and unclear.

- L127: “were” instead of “was”?!

- L141-144: Can you describe the purpose and the chosen threshold in a bit more detail?

- L154: What is SIMPER?

- L190-191: How is this taking into account the different mound locations?

Validity of the findings

While some of the descriptive findings are certainly valid and meaningful, I do have some concerns about the details of some of the analyses and the conclusions drawn given a low sample size, but also the details of some of the analyses and my current lack of understanding some of the aspects.

- What I had generally really hoped to see was a statistical comparison also of the different samples taken from the same sample type. The authors did take multiple samples from the same location, but, unless I missed it, neither analysed that nor specified what the purpose was otherwise. This might have perhaps been limited by the samples size, which is unfortunately rather low – but can still provide useful insides, I believe. Nevertheless, I would find the results presented in the manuscript, that basically all the microbial communities differed between categories and locations much more convincing if it was contrasted with something where we would expect a higher similarity, e.g. the difference within shallow soil samples. How the analysis was done, we do not really know what baseline level of similarity we would expect for samples from the similar source,

- L151-153: Did you assess the fit of the data in the nMDS? What were the associated stress values?

- L153-154/Fig.1: Is this correct, it seems that many of the phyla shown actually have a percentage of less than 1%. Also it’s very hard to make out which bacterial phyla the much more frequent sequences belong to, as there are too many similar colour shades and I’m also not able to count to order cause many of the bars are so low and the background grid doesn’t match the width of the bars. Maybe they could at least be numbered on the y axis. Otherwise, I’m not sure what information I am supposed to get from this figure.

- - Fig.2: Why does the figure show unrarefied data? It seems however, that the samples that were described to have been removed in L141-144 are not included.

- L163-164/Table 1: Is this averaged over the different samples collected from the different sites? This should be mentioned. Generally, the table might also be a good spot to add the number of samples per each site that were actually used in the analyses. Does Faeces B state an SD of 0 because it is only one sample? I do not think that it makes any sense to calculate an SD from one sample; this is misleading.

- Given above mentioned concerns, limitations in sample size and statistical power, and the descriptive nature of the study, I would tune down some of the wording used in the discussion section to more careful interpretations. After the authors eagerly reject all hypothesis as for the origin of the bacteria, they do not present an alternative explanation for their source. Thus, I would recommend being a bit more careful with rejecting hypothesis and jumping to strong conclusions and give a bit more nuanced picture of what we can really say from the presented results.

Additional comments

I generally like the study, but the analyses lack clarity for me in the current form and I would not want to draw the presented conclusion from the data as it is presented.

---

## Round 0.2 · accepted · Accept

Your manuscript has now been reassessed by two of the original reviewers. Both agree that all requested changes to the manuscript have been made in a satisfactory manner. I am pleased to recommend acceptance of your manuscript.

·

Basic reporting

This is a revision of the first version. The Authors have done an excellent job at addressing the comments raised by the editor and reviewers. I recommend acceptance for publication.
I noticed one minor typo no line 243: ‘significantly’
The article is well written, the English is clear and concise, the appropriate literature is cited, figures and tables are appropriate, hypotheses are well stated and tested.

Experimental design

Experiential and statistical design is appropriate.

Validity of the findings

The analysis and interpretation of data is appropriate and the conclusions made are well supported.

Additional comments

This is a revision of the first version. The Authors have done an excellent job at addressing the comments raised by the editor and reviewers. I recommend acceptance for publication.
I noticed one minor typo no line 243: ‘significantly’

Reviewer 2 ·

Basic reporting

The article is clear and well written. I think the manuscript has greatly improved from the previous version I have reviewed and the authors have addressed all my former points sufficiently.

I really enjoyed the additional information that was added on the ecology and behaviour surrounding the use of the mound. E.g., I previously hadn’t been aware that those mounds could exist for years or even decades and I think it’s an important detail to understand the distinct bacterial community within the mounds.

Experimental design

As stated in my previous review of this work, I think the experiment is sound and any concerns I had about regarding methodological aspects and data analysis have been addressed and corrected already.

Validity of the findings

All data have been provided, as far as I can see.

I appreciate the adjustments the authors have made during the revisions regarding the additional analysis and not overstating the conclusions that can be drawn.

Additional comments

I have no addititional comments.